# Efficacy of the Nutrition Education and Screening Tool as a Foundation for Exploring Perinatal Diet and Determinants in Aboriginal and Torres Strait Islander Women of Far North Queensland

**DOI:** 10.3390/nu16193362

**Published:** 2024-10-03

**Authors:** Janelle James, Karen Yates, Cate Nagle

**Affiliations:** 1College of Healthcare Sciences, James Cook University, Townsville, QLD 4811, Australia; 2School of Pharmacy and Medical Science, Griffith University, Southport, QLD 4215, Australia; 3Centre for Quality and Patient Safety Research, Deakin University, Burwood, VIC 3125, Australia

**Keywords:** Aboriginal and Torres Strait Islander, Indigenous, pregnancy, food frequency, nutrition assessment, diet

## Abstract

Background/objectives: Assessing perinatal diet and its determinants in Australia’s Aboriginal and Torres Strait Islander women remains challenging, given the paucity of tools that incorporate Aboriginal ways of knowing, being, and remembering within a quantitative framework. This study aimed to explore the determinants of perinatal nutrition in this population and to evaluate the efficacy of the Nutrition Education and Screening Tool (NEST) in collecting diet-related data in this population. Methods: This study employed a Participatory Action Research approach using the NEST as a foundation for structured research inquiry. Self-reported diet and determinants were collected from a cross-sectional cohort of Aboriginal and Torres Strait Islander women from Far North Queensland. Results: Participants (*n* = 30) declared excess consumption of meat and alternatives, fruit, vegetables and legumes, and dairy and alternatives. Grain and cereal consumption aligned with recommendations; wild-harvested foods comprised a mean 19.75% of their protein intake. Food frequency data were supported by participants’ descriptions of how they eat, combine, rotate, and cook these foods. Conclusions: Standard food frequency questionnaires are challenging for Aboriginal and Torres Strait Islanders as their concepts of time and ways of remembering are different from Western understanding. Use of the NEST allowed food frequency items to be explored, clarified, and cross-referenced; yarning provided a degree of support for quantitative data. The results of this study translate to future public health research, practice, and policy. Alternative quantitative measures to determine food frequency should be considered in future studies. These may include the cyclical approach to time that is well understood and integrated by Indigenous cultures.

## 1. Introduction

According to the 2021 census, the Cape York and Torres Strait regions were home to approximately 15,000 people identifying as Aboriginal and Torres Strait Islander; 23% were women of reproductive age (15–44 years, *n* = 3427) [1]. Many of these women experience challenges in commencing and maintaining a healthy pregnancy, including over- and under-nutrition-related disorders [2]. Maternal malnutrition influences perinatal outcomes [3], potentially resulting in foetal growth restriction, reduced physical readiness for birth, suboptimal postnatal recovery, and reduced rates of breastfeeding establishment and continuation, thereby impacting transgenerational health.

Assessing, developing, implementing, and evaluating nutrition programs in remote Indigenous communities is challenging; factors including geography, English literacy, resourcing, and the effects of colonisation can be barriers to conducting research [4,5]. As such, dietary intake cannot be examined in isolation. Food choices, the factors, and the motivations driving them must also be assessed to develop a foundation for strategies that enhance outcomes and result in effective culturally responsive strategies [6].

While strategic nutrition research remains a priority in Far North Queensland (FNQ) [7], the instruments available to assess dietary intake, quality, and influences during pregnancy remain unvalidated in Aboriginal and Torres Strait Islander populations. Further, pen-and-paper food frequency questionnaires are impractical given the barriers described [5]. Therefore, the validation of a culturally appropriate digital method of data collection is an important step towards consistent nutrition assessment, identifying supportive influences and modifiable behaviours, and the development of strengths-based strategies to address known health inequities in pregnant women of FNQ.

Featuring minimal language and easily recognisable images, the Nutrition Education and Screening Tool (NEST) is a novel digital nutrition assessment instrument adapted from the Norwegian Mother and Child (MoBa) Study [8], Harvard [9], the Block [10], and diet history questionnaires [11] according to the methods described by Cade et al. [12]. The NEST Food Frequency Questionnaires (FFQ) facilitate the declaration of 287 individual foods across eight consecutive surveys (meat and other protein; bush tucker; grains and alternatives; fruit and vegetables; dairy foods; hot and cold drinks; takeaway and junk food; and spreads, oils, sauces, and dressings), and intake calculations for 231 unique food constituents over the preceding four-week period using consistent frequency options (2+ per day, once a day, 2–4 times per week, once a week, 2–4 times per month, once a month, and never); the food groups with higher daily recommendations have slightly different upper range frequency options (2–3 per day, 4–5 per day, and 6+ per day).

The NEST Food Choice Questionnaire (FCQ), adapted for research in pregnancy from Steptoe, Pollard, and Wardle’s Food Choice Questionnaire [13], also supports respondents to select their food choice motivations with 22 motives available for concurrent selection. These motives are further organised into nine food choice factors—health, mood, convenience, sensory appeal, natural content, price, weight control, familiarity, and pregnancy factors. This survey series can be ‘nested’ between additional instruments to collect complementary data and has the flexibility to incorporate co-design with target populations, facilitating the examination of relationships between diet, dietary determinants, and additional desirable data in varied contexts. Taking advantage of digital technology, the NEST has proven efficacy in pregnant women, successfully describing habitual diet and its determinants in culturally diverse populations in South-East Queensland [14,15,16,17,18,19].

Given the challenges and health inequities experienced by many Aboriginal and Torres Strait Islander communities, the NEST may offer an alternative way of assessing and understanding ‘nutritional behaviours’; the planned, spontaneous, or habitual actions of individuals to procure, prepare, and consume food, and the holistic health implications of nutrition and its determinants during pregnancy [20].

The Mookai Rosie Mums and Bubs project was undertaken in an Aboriginal Community Controlled Health Organisation (ACCHO) in Far North Queensland. This co-designed project aimed to gain a deeper understanding of the determinants of perinatal nutrition in Aboriginal and Torres Strait Islander women from remote Far North Queensland communities. Further, we aimed to evaluate the efficacy of the NEST as a foundation for exploring perinatal diet and determinants in this cohort. This paper reports the quantitative findings of this research; qualitative findings have been reported elsewhere.

## 2. Materials and Methods

### 2.1. Study Design and Setting

To enhance cross-cultural understanding and utilise co-design processes, both Indigenous and western research methods were applied. Honouring Aboriginal and Torres Strait Islander ways of viewing, creating, and experiencing their world was central to this research; however, the inclusion of western methods [21,22,23,24,25] was necessary to examine points of parallel and difference [26]. The study used a Participatory Action Research (PAR) approach using the NEST as a foundation for information sharing and structured research inquiry. The food frequency surveys in the NEST provided a loose but consistent framework, assisting participants to stay on topic as well as offering opportunities to expand through yarning at arising points of interest. Quantitative data were collected using a cross-sectional survey design to determine participants’ self-reported dietary recall and determinants. During an individual yarning session, the researcher and participant together progressed through each of the NEST food frequency surveys, systematically addressing each of the individual food and frequency options. This data declaration was complemented by a 24 h recall and clarification through iterative yarning.

### 2.2. Participants

Participants were pregnant Aboriginal or Torres Strait Islander women from communities in remote Far North Queensland who had transferred to a tertiary health service for intrapartum care (Cairns). Thirty participants aged 16–44 years from Cape York and Torres Strait communities receiving maternity care from the ACCHO were recruited opportunistically; all pregnant women who had the capacity to consent were invited to participate (*n* = 42, 71% participation rate). Eligibility within this cohort was limited by language due to the requirements for informed consent and the number of communities serviced; however, Creole and English are common throughout the region. Creole-speaking prospective participants were supported by team members with Creole fluency; no further inclusion or exclusion criteria were applied.

### 2.3. Recruitment and Consent

The recruitment of women occurred between 18 February and 12 September 2022. The Research Assistant (RA) provided the study information and, where necessary, provided support with language and understanding prior to seeking consent; data collection commenced after consent was confirmed. As the RA was not directly responsible for the provision of health care, no conflict of interest was identified. Participants were informed that their participation was voluntary and that they could withdraw from the study at any time without any effect on their care. Participants completing the data collection were given a $50 voucher from a local Independent Grocers of Australia in accordance with the NHMRC guidelines for payment of research participants [27].

### 2.4. Data Collection

Dietary data were collected and managed using the NEST and uploaded to the Research Electronic Data Capture (REDCap) application hosted on the James Cook University server [28]. REDCap is a secure, web-based software platform designed to support data capture for research studies, providing modifiable user rights, identifiable data protection, an intuitive interface, audit trails, automated export, and procedures for data integration and interoperability with external sources [29]. The researchers sat with the participants and completed the NEST FFQ and FCQ surveys on the computer; each participant was therefore free to yarn without technical distractions. Completed dietary data were downloaded and entered into the NEST calculation tables to produce data relating to energy and individual nutrient intake. Participants engaging in the 24 h dietary recall detailed their meals (breakfast, lunch, dinner, morning, and afternoon tea) in the 24 h preceding the data collection; the reported diet was allocated into the five main food groups manually. Additionally, meals were manually allocated into three categories—no meal, healthy food, and unhealthy food. Information relating to the participant’s health and pregnancy, relevant pathology results (blood examination, iron studies, and glycaemic screening), and birth outcomes (pregnancy complications, gestational weight gain, gestation, onset, mode and complications of birth, infant anthropometrics, sex, and length of stay for the mother and baby) were obtained directly from hospital records with the participant’s consent.

### 2.5. Data Analysis

Descriptive statistics were used to summarise demographics, anthropometric, and biochemical measures (*n*, %). Continuous variables were assessed for normality; data were normally distributed where skewness and kurtosis were between −3.0 and +3.0 and the Shapiro–Wilk value was not statistically significant. Normally distributed data have been described using mean, standard deviation, and 95% confidence interval (mean [±SD, 95% CI]); variables that were not normally distributed were represented with median, range, and interquartile range (IQR). Pearson’s product-moment correlation test (*r*, *p* value), chi-square analyses (χ^2^, *df*, *p* value), and one-way ANOVA (β, 95% CI, *p* value) were used to identify and explore relationships between normally distributed variables; abnormally distributed variables were examined using Spearman’s Rho (*r*, *p* value) and Wilcoxon Signed Rank test (*z*, *p* value). Binary logistic regression was used to further examine identified associations between categorical variables (OR, 95% CI, *p* value). Data were analysed using SPSS v29.

### 2.6. Research Team

The research team was composed of both Aboriginal and Torres Strait Islander and non-Indigenous stakeholders, who worked together to co-conceptualise, co-design, co-implement, and co-evaluate this project. The non-Indigenous principal investigator (PI) proposed the project, obtained grant funding for the conduct of the research, and facilitated use of the NEST. The organisation’s Chief Executive Officer, Health Program Manager, a senior Aboriginal researcher, and the Aboriginal and Torres Strait Islander Health workforce collaborated with the PI to modify and implement the NEST and integrate yarning opportunities. Health workers worked alongside the PI during data collection. The PI and health workers led the data collection; however, each participant exercised autonomy in how the data collection progressed, expanding on priority topics through yarning. The data collected are therefore a reflection of the priorities of participants. Additionally, Indigenous administrative staff were trained as RAs and were responsible for the research introduction and consenting process with potential participants.

### 2.7. Research Team Training

During the development phase of this project, meetings between the PI and the ACCHO were conducted weekly. During these meetings, knowledge-sharing occurred between all parties regarding the service, their research and development goals, the creation of a research protocol, ethics, governance, and funding applications. After these processes, a research training schedule was developed with the needs of the staff and service in mind. This research training included the aims and objectives of the project, orientation to the NEST, foundations of research, and research yarning theory. This training was conducted by the PI and the senior Aboriginal researcher. The RAs were trained in the informed consent process and database administration. Staff were supported in the conduct of the research, and regular feedback and quality improvement sessions were conducted with the research team.

### 2.8. Privacy and Safety

Participants were assigned a study number on consent and asked to provide enough information to enable identification of their previously described pathology results and birth outcomes via processes required by Queensland Health (i.e., full name/DOB/Medical Record Number). Dietary, health, and demographic declarations were entered directly into the database; results gleaned from hospital records were uploaded into the participant study number file. Any identifiable data provided by participants were kept separate from their declared data in a secured folder, ensuring privacy and confidentiality as per the university and health service data privacy policies. Hard copies of consents and transcripts were shredded after scanning; scanned documents and declared data were stored in the secure JCU REDCap database under password protection. Raw voice files were deleted from the recording device after transcription. The PI entered pathology results and birth outcomes provided by the health service into each participant’s REDCap ID. Data obtained from hospital records was re-identified with the participant study number on entry to the project database. On completion of the data entry, all records were de-identified. Participants were advised by the health worker that all information remained confidential within the recruiter’s duty of care and mandatory reporting responsibilities. Women were advised they may choose not to answer any questions that made them feel uncomfortable.

## 3. Results

A total of 30 women participated in the study, representing 14 unique communities (Figure 1).

All participants provided quantitative data; data that were assessed to be unrealistic were excluded from the analysis. The majority of participants identified as Aboriginal (66.7%); two-thirds declared multiparity (63.3%), and two-thirds smoked tobacco (63.3%). Two-thirds of the pregnancies were unplanned. Outreach participants comprised 40% of participants (*n* = 12); 17 participants were in their final weeks of pregnancy at the time of data collection (56.7%), with a further 13 (43.3%) participating in the early postnatal period (7–10 days). Demographic, anthropometric, and biochemical features of the participants have been summarised in Table 1.

### 3.1. Energy Intake

Mean energy intake was calculated from the declared habitual diet using the NEST calculation tables. The calculated mean energy intake of the participants was 14,586 kJ/day (±SD 4888 kJ/day), a figure within the recommended daily energy requirements of pregnant women (Table 2) [30]. Mean daily energy intake was correlated with food selection motives ‘budget’ (*r* = 0.388, *p* = 0.041) and ‘ease of availability’ (*r* = 0.412, *p* = 0.029); significant correlations were found between energy intake and the number of serves of each food group per day, an expected outcome given energy calculations were derived from the food frequency declarations. Energy intake was not associated with any demographic, health, or outcome variables.

### 3.2. Food Groups

Participants declared their usual diet on Country; fruit and dairy intake demonstrated abnormal distribution. Intake was calculated to be higher than recommendations for pregnant women for four of the five core food groups (19–50 years, Table 2)—meat and alternatives (6.39 [±2.64]), fruit (5 [2–11]), vegetables and legumes (7.07 [±2.23]), and dairy and alternatives (3 [1–7]). Grains and cereal consumption aligned with recommendations (8.46 [±2.27]) [31]. Wild-harvested foods comprised 19.75% (±14.74) of their protein intake; whole grains constituted 31.79% (±22.21) of the declared grain and cereal intake.

### 3.3. 24 H Recall

Twenty-seven participants completed the 24 h dietary recall. The most frequently eaten meal was dinner (*n* = 27). Twenty participants consumed a healthy evening meal (74.1%); 25.9% had an unhealthy meal. Breakfast was consumed the least; six participants (22.2%) stated they had not eaten that morning, with a further three declaring unhealthy morning food choices. Serves per day of each food group across the 24 h recall achieved normal distribution except for fruit (0.0 [0–0]). The difference between 24 h and their declared habitual fruit intake reached statistical significance (*z* = −4.209, *p* < 0.001) with a large effect size (*r* = 0.825). Intake of grains and cereals (4.1 [±1.6]), vegetables and legumes (1.26 [±1.22]), and dairy and alternatives (0.85 [±1.23]) also fell well short of recommended daily intake; they were also lower than declared habitual intake. Participants approached the recommended daily serves of meat and alternatives (3.19 [±1.60]) during the 24 h period [31].

### 3.4. Food Choice

Participants most frequently declared ‘tastes good’ as a motive for food choice (*n* = 25, 83.3%); the least declared motives were ‘calorie’, ‘fat’, ‘fibre’ (*n* = 3, 10%), and ‘preservative content’ (*n* = 1, 3.3%). Familiarity was a consistent theme, with the five ‘familiarity’ motives each being selected by 40–60% of the participants (habit [43.3%], it is familiar [50.0%], cultural beliefs [60.0%], friends and family eat it [60.0%], and childhood memories [40.0%]; ‘price’ (budget [66.7%] and value for money [53.3%]) were also a constant factor, as was ‘sensory appeal’ (tastes good [83.3%], looks good [50.0%], and smells good [63.3%]). While 50.0% of the cohort declared selecting food due to ease of availability, ease of preparation was only a consideration for 26.7% of participants (Table 3).

Participants whose food choices were influenced by cultural beliefs birthed infants with a higher mean birthweight than those whose were not (3467 vs. 3054 g, *p* = 0.012). Only pre-pregnancy BMI retained statistical significance in the multiple regression models. Neither Indigenous identity nor annual income was significantly associated with 24 h recall, habitual food frequency, or food selection motives. Glycaemic control was influenced by several factors. Fasting glucose decreased in participants choosing foods aligning with those consumed by friends and family (β = −0.975, 95% CI −1.705, −0.245, *p* = 0.017); this decrease was also observed with increasing serves of meat and alternatives (β = −0.234, 95% CI −0.437, −0.031, *p* = 0.031).

Significant correlations were found between food groups and selection motives. The percentage of dietary whole grains was positively correlated with ‘health’ factor motives (‘gives me energy’ [*r* = 0.376, *p* = 0.048], and ‘it is good for me’ [*r* = 0.485, *p* = 0.009]), in addition to pregnancy-related motives; however, diabetes in pregnancy was not a factor in this decision (*p* = 0.828). ‘Looks good’ was a selection motive significantly and positively correlated with fruit (*r* = 0.504, *p* = 0.009), vegetable (*r* = 0.472, *p* = 0.013), and dairy (*r* = 0.666, *p* < 0.001) intake; the percentage of wild-harvested protein was correlated with taste (*r* = 0.381, *p* = 0.046). This aligns with the association between the intake of meat and alternatives and ‘cultural beliefs’ as a food choice motive (*r* = 0.466, *p* = 0.025). Intake of grains and cereals was also correlated with selection based on cultural norms (*r* = 0.407, *p* = 0.031). Fruit (*r* = 0.467, *p* = 0.016) and meat intake (*r* = 0.443, *p* = 0.023) were positively correlated with perceived nutrition value.

## 4. Discussion

The participants in this study described a high-protein diet, nearly all of which was sourced from animals. This preference for animal proteins continued when they were transferred out of the community, with the 24 h recall demonstrating a focus on acquiring and consuming red meat and chicken. Fish and seafood constituted a large proportion of animal proteins in the community; this changed when away from home due to a stated inability to apply traditional harvesting techniques [32]. This cohort declared an intake of fruit and vegetables well in excess of those reported in previous research [33,34,35,36]; participants described a few barriers to accessing fresh fruit and vegetables, in addition to detailing alternative options when the community shop supply was affected. While the accuracy of food frequency declarations may be debatable, the ability of each participant to independently describe how they eat, combine, rotate, and cook these foods supports their frequency declarations; further, the alignment between data from different participants and communities lends credence to these findings.

Participants expressed a preference for wild-harvested protein sources, in particular seafood, the harvesting of which was also described as being central to social and cultural practices [32,37]. This preference was also supported statistically, with the percentage of wild-harvested protein correlating with ‘taste’ as a motivation for food choice and the association between the intake of meat and alternatives with cultural beliefs [38]. Women in this study expressed the casual and frequent nature of seafood gathering as well as the different methods of cooking these wild-harvested foods. These methods—one pot and fire-based, such as the Kup Murri—were often simple but time-consuming and represented lengthy dedication to food preparation. However, this time was considered valuable learning, teaching, and connection time, enhancing social capital and ensuring a significant long-term return on investment [4]. This aligned with the emphasis on ease of availability over preparation time and the proportion of the cohort reporting familiarity-based motives for food choice.

Researchers were able to gather information from participants using the NEST, applying the surveys as both a quantitative data collection instrument and a foundation for yarning about food during pregnancy. Further, use of the NEST assisted to lay the foundation for describing a sociological construct with complex inherent interrelatedness, community dynamics, and nutritional behaviours [39]. The calculated mean energy intake of the cohort was found to be higher than recommendations for late pregnancy and early postpartum; however, this finding is similar to those published by Lee et al. in 2016 [40]. While this calculation is likely confounded by the difficulty with declarations of dietary intake across time, it is supported by the mean pre-pregnancy BMI demonstrated by the cohort and the participants’ description of how and what they eat, traditional food abundance, and food security strategies. This calculation was also well supported by consistent declaration of these nutrition behaviours across most of the cohort, irrespective of their home community; qualitative data supported quantitative results [41].

Habitual dietary intake was calculated to be higher than recommendations for four major food groups, and this contributed to the mean energy consumption sitting at the high end of the recommended energy intake range. However, these too were aligned with previous research [40] and were further supported by the qualitative data, with women describing in detail their food combinations and how their dietary behaviours were situated in the social context. Therefore, the data declared is considered to be an accurate representation of the participants’ experience of dietary intake and their nutritional behaviours. The conduct of this study was well received and supported by the participants; the mean yarning session time of 61 min aligned with the participant and organisational resource allocation, and the data collection requirements. Additionally, participant feedback regarding their experience of the interview and their engagement with the contemporaneous results review indicated satisfaction and curiosity creation, enhancing their nutrition literacy [42].

In this context, and under these data collection circumstances, the NEST has demonstrated findings that represent the truth of the cohort. This truth is supported statistically by low standard deviations for food group calculations, energy intake calculations, anthropometric measures, and participants’ perspectives. As such, the NEST has demonstrated broad internal and external validity, providing a foundation for further validation studies in this population [43].

### Strengths and Limitations

The conduct of this study experienced several challenges that required flexibility and responsiveness to overcome. However, the design of the NEST supported a degree of fluidity to address research barriers unique to the cohort. The co-design of this project was a strength, as was the partnership between the Indigenous and non-Indigenous team members. The broad range of communities represented supports generalisability for Aboriginal and Torres Strait Islander communities, but low numbers for distinct communities limit the findings for specific subcultures. Similarly, the total number of participants resulted in low statistical power, limiting the conduct of additional data analysis.

The ratio of antenatal to postnatal participants has facilitated the collection of nutrition-related data across the perinatal period and added a layer of complexity concerning mobility and access to foods. Physical recovery, the demands of caring for a newborn, accommodation providers, social supports, and transport in the regional centre may influence food security and accessibility factors, an aspect that was not explored in this study. Participants had been away from their home community for variable lengths of time at the point of data collection. This may have been due to pregnancy complexity, timing of birth, participation in the antenatal or postnatal period, or the opportunistic and cross-sectional nature of recruitment.

As the usual dietary patterns of participants changed significantly at the time of transfer out of the community, modifications needed to be made to the administration of the FFQ. As such, rather than collecting data related to a four-week time period, questions were reframed to inquire about dietary intake and habits when in their home communities versus since they arrived in Cairns. Some participants demonstrated strong emotional connections to certain foods, particularly meat and other proteins. This affected the quality of the data collected, as their longing for the food was confused with the reality of intake. This resulted in the exclusion of some data. The difference between participants’ habitual and Cairns-based dietary intake also prevented validation of the FFQ using 24 h recall as a comparative measure.

The application of standard food frequency options created challenges for this cohort, as they did not align with Aboriginal and Torres Strait Islander concepts of time [44] or ways of remembering [45]. Food frequency declarations often required exploration, clarification, and cross-referencing. However, the final calculations are supported by the qualitative data gathered during the yarning session. While yarning topics provided a degree of support for quantitative data, consideration should be given to the use of alternative quantitative measures to determine food frequency in future studies [38]; these should include the cyclical approach to time that is well understood and integrated by Indigenous cultures [44,46].

## 5. Conclusions

Standard food frequency questionnaires are challenging for Aboriginal and Torres Strait Islanders as their concepts of time and ways of remembering are different from Western understanding. Use of the NEST allowed food frequency and nutrition behaviours to be explored, clarified, and cross-referenced; yarning topics provided a valuable degree of support and clarity for quantitative data. As such, the accuracy of food frequency data in this population would be enhanced by using a Participatory Action Research rather than a standardised instrument and quantitative approach. These findings support the use of the NEST in Indigenous nutrition research practice. However, the validity of results is dependent on research design, flexibility, rapport, and the ability to cross-reference quantitative and qualitative data. The benefits of the structure and content of the NEST rely on a culturally sensitive approach to all facets of research.

The results of this study translate to future public health research, practice, and policy. Future research and practice can be informed by using the NEST, a tested instrument that encompasses Aboriginal and Torres Strait Islander concepts of time, providing a more accurate measure of nutritional intake. Policy-makers have long been aware of the adverse impacts of women not birthing on Country. These results complement previous research and highlight that the move away from country forces fundamental changes to the woman’s nutritional intake.

## Figures and Tables

**Figure 1 nutrients-16-03362-f001:**
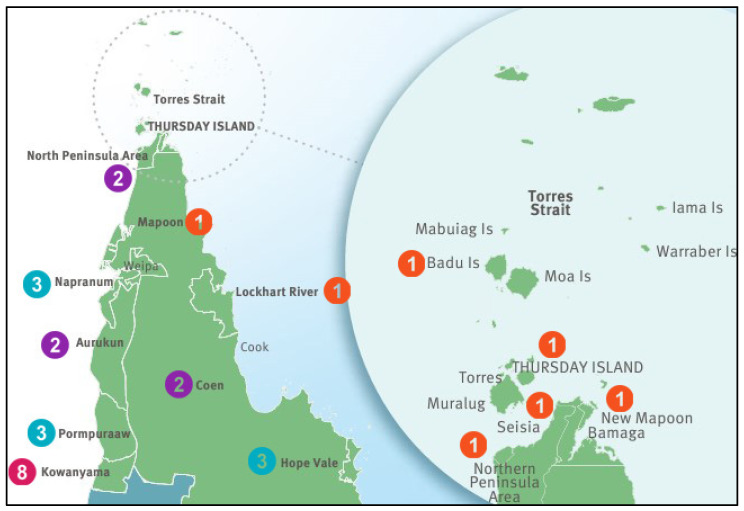
Number of participants by community.

**Table 1 nutrients-16-03362-t001:** Descriptive characteristics of participants.

Variable	Variable	*n* (%)
Community	Aurukun	2 (6.7)
	Badu Island	1 (3.3)
	Coen	2 (6.7)
	Hope Vale	3 (10.0)
	Injinoo	2 (6.7)
	Kowanyama	8 (26.7)
	Lockhart River	1 (3.3)
	Mapoon	1 (3.3)
	Napranum	3 (10.0)
	Old Mapoon	1 (3.3)
	Pormpuraaw	3 (10.0)
	Seisia	1 (3.3)
	Thursday Island	1 (3.3)
	Umacigo	1 (3.3)
Indigenous identity	Aboriginal	20 (66.7)
	Torres Strait Islander	4 (13.3)
	Both Aboriginal and Torres Strait Islander	6 (20.0)
Education	Did not attend high school	3 (10.0)
	Did not finish high school	11 (36.7)
	Finished high school	8 (26.7)
	Technical and Further Education	8 (26.7)
Estimated net annual family income	Less than $30,000	16 (53.3)
	$30–$50,000	7 (23.3)
	$50–$70,000	4 (13.3)
	$90–$120,000	1 (3.3)
	Not answered	2 (6.7)
Parity	Nulliparous	11 (36.7)
	Multiparous	19 (63.3)
Planned pregnancy	Yes	11 (36.7)
	No	19 (63.3)
Smoke cigarettes	Yes	19 (63.3)
	No	10 (36.7)
	Declined to answer	1 (3.3)
Diabetes in pregnancy	Yes (gestational diabetes)	7 (23.3)
	Yes (Type II)	2 (6.7)
	No	20 (69.0)
	Unknown	1 (3.3)
Maternal age (years, mean ± SD)		27 (±7)
Pre-pregnancy BMI (kg/m^2^, mean ± SD)		25.8 (±6.0)
Haemoglobin (g/L)	With iron infusion (*n* = 8)	117.1 (±19.86)
	Without iron infusion (*n* = 22)	117.3 (±9.21)
Oral glucose tolerance test (75 g load) values (mg/dL)	Fasting (*n* = 8)	4.6 (±0.65)
	1 h (*n* = 8)	8.4 (±2.11)
	2 h (*n* = 8)	7.2 (±2.00)
Haemoglobin A1c (*n* = 19)		5.4 (±0.65)
Infant birthweight (grams, mean ± SD)		3305 (±479)

**Table 2 nutrients-16-03362-t002:** Statistical analysis of declared dietary intake.

	*n*	Shapiro–Wilk (*p* Value)	Mean (SD)	Median (Range)	95% CI	IQR	RDI (Serves)
Habitual intake							
Meat and alternatives (serves/day)	23	0.232	6.39 (2.641)	n/a	5.25–7.53	n/a	3.5
Grains and cereals (serves/day)	28	0.548	8.46 (2.269)	n/a	7.58–9.34	n/a	8.5
Vegetables and legumes (serves/day)	27	0.258	7.07 (2.235)	n/a	6.19–7.96	n/a	5
Fruit (serves/day)	26	0.04 *	n/a	5 (2–11)	n/a	3	2
Dairy (serves/day)	28	0.008 *	n/a	3 (1–7)	n/a	2	2.5
Wild-harvested protein (%)	23	0.970	19.75 (14.744)	n/a	14.03–25.47	n/a	n/a
Whole grains (%)	28	0.060	31.79 (22.210)	n/a	23.17–40.40	n/a	50
Energy intake (kJ/day)	22	0.554	14,586 (4888)	n/a	7197–25,818	n/a	7200–14,600
24 H recall							
Meat and alternatives (serves/day)	27	0.008 *	n/a	3 (0–6)	n/a	2	3.5
Grains and cereals (serves/day)	27	0.001 *	n/a	3 (2–10)	n/a	3	8.5
Vegetables and legumes (serves/day)	27	<0.001 *	n/a	1 (0–4)	n/a	2	5
Fruit (serves/day)	27	<0.001 *	n/a	0 (0–4)	n/a	0	2
Dairy (serves/day)	27	<0.001 *	n/a	0 (0–4)	n/a	2	2.5

* Statistically significant.

**Table 3 nutrients-16-03362-t003:** Food choice factors and motives (food choice factors are not mutually exclusive; participants could select more than one response).

Factor	NEST FCQ Options	*n* (%)
Factor 1—health	Gives me energy	11 (36.7)
	Nutritious	12 (40.0)
	It is good for me	16 (53.3)
	Fibre content	3 (10.0)
Factor 2—mood	Comfort food	13 (43.3)
Factor 3—convenience	Ease of preparation	8 (26.7)
	It is easily available	15 (50.0)
Factor 4—sensory appeal	Tastes good	25 (83.3)
	Looks good	15 (50.0)
	Smells good	19 (63.3)
	Texture	5 (16.7)
Factor 5—natural content	Preservative content	1 (3.3)
Factor 6—price	My budget	20 (66.7)
	Value for money	16 (53.3)
Factor 7—weight control	Calorie content	3 (10)
	Fat content	3 (10)
Factor 8—familiarity	Habit	13 (43.3)
	It is familiar	15 (50.0)
	Cultural/ethnic beliefs	18 (60.0)
	My friends and family eat it	18 (60.0)
	Childhood memories	12 (40.0)
Factor 9—pregnancy	Cravings and taboos	19 (63.3)

## Data Availability

The raw data supporting the conclusions of this article will be made available by the authors on request.

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
