# Peer review of "Efficacy of the Nutrition Education and Screening Tool as a Foundation for Exploring Perinatal Diet and Determinants in Aboriginal and Torres Strait Islander Women of Far North Queensland"

_nutrients, 2024, doi:10.3390/nu16193362_

Round 1

Reviewer 1 Report

Comments and Suggestions for Authors

I read with interest the paper titled "Efficacy of the Nutrition Education and Screening Tool as a foundation for exploring perinatal diet and determinants in Aboriginal and Torres Strait Islander women of Far North Queensland."

I have minor comments to add. 

1. In background, you referred iron deficiency; however report an increased consumption of meet. Is this not contradictory?

2. How many women were invited to participate? Given this number, whats the response rate?

3. Methods: The Shapiro–Wilk test had been more appropriate method for small sample size (<50 samples), which is the case. What are the results of Kolmogorov-Smirnov tests for each variable tested? 

4. Excluding outliers to get normal data is a poor and biased practice. Non-parametric tests should be used instead. 

5. Whats the adjustment that were made to get normal data? 

6. Was Pearson correlation used when data is not normal? Why not use Spearman rho instead?

7. ANOVA alternative should be used when data was not normal. 

8. Table 1 - GDM should be defined. 

9. Please consider to add a footnote on Table 2 stating that items are non-mutually exclusive and multiple choices could be taken. 

10. The information on sections 3.1 and 3.2 is hard to read. I suggest to put that in a plot or table comparing the consumption with the reccomended amounts. 

Author Response

  1. In background, you referred iron deficiency; however report an increased consumption of meet. Is this not contradictory?

Thankyou, the authors agree that this is a perplexing situation. The women in this cohort did not demonstrate IDA (per the haemoglobin pathology), however there is a there is a body of research that highlights the increased incidence of IDA in this population. The reasons for this misalignment are unknown and warrant further exploration beyond the scope of this paper. The wording has been changed to eliminate the contradiction (line 37-38).

  1. How many women were invited to participate? Given this number, whats the response rate?

This has been added into the participants section of the methods (line 115)

  1. Methods: The Shapiro–Wilk test had been more appropriate method for small sample size (<50 samples), which is the case.
    The data has been re-examined using the Shapiro-Wilk test; this has been stated in the manuscript (line153).
  2. What are the results of Kolmogorov-Smirnov tests for each variable tested?

The results of the Shapiro-Wilk test have been included in Table 2, thank you.

  1. Excluding outliers to get normal data is a poor and biased practice.
    The data was examined for outliers to determine whether the data lay within the realms of possibility rather than to normalise the data. Data that was grossly overstated due to the participants emotional or cultural connection to the food has now been excluded from the analysis. The statement regarding outliers has been amended to reflect this (line 153, 222). This has also been added into the limitations section (lines 367-370).
  2. Non-parametric tests should be used instead. 

Thank you, non-parametric tests have now been used to analyse the dataset where appropriate. This has been detailed in table two (page 7) and in the methods (lines 156-159).

  1. Whats the adjustment that were made to get normal data? 

Unrealistic data was not included in the analysis for each element. The number of responses included in each element has been included in Table 2.

  1. Was Pearson correlation used when data is not normal? Why not use Spearman rho instead?

Thank you for highlighting this error. Abnormally distributed variables have been re-analysed with the correct statistical method. This has been amended in the manuscript on lines 157-8, 277 and 282.

  1. ANOVA alternative should be used when data was not normal. 

Thank you for this prompt. Abnormally distributed data was reanalysed using the Wilcoxon Signed Rank test. This has been noted in the methods (line) and the results (lines 247-9).

  1. Table 1 - GDM should be defined. 

Thank you for pointing this out, this has been defined within the table.

  1. Please consider to add a footnote on Table 2 stating that items are non-mutually exclusive and multiple choices could be taken. 

Thankyou for this suggestion; the footnote has been added at the bottom of page 7

  1. The information on sections 3.1 and 3.2 is hard to read. I suggest to put that in a plot or table comparing the consumption with the reccomended amounts. 

Your suggestion is appreciated. Some of this data has been removed from this section and added into a dedicated table (Table 2, page 7). Data relating to 24 hour recall has also been added into this table to make comparison between the two timepoints more straightforward.

Reviewer 2 Report

Comments and Suggestions for Authors

Janelle James et al. submitted to Nutrients an article, dealing with the efficacy of NEST as a foundation for exploring perinatal diet and determinants in a local area.

Since it is not stated in the materials and methods, it is unclear whether this study was conducted “from January and June 2022". If so, this study is out of date, having been conducted more than two years ago.

The Authors must explain how this work provides useful ideas for improvement related to the topic discussed, also in terms of Public Health and decisions that could be advanced by policy makers.

Comments on the Quality of English Language

Minor editing of English language required.

Author Response

1. Since it is not stated in the materials and methods, it is unclear whether this study was conducted “from January and June 2022". If so, this study is out of date, having been conducted more than two years ago.

Thank you for the opportunity to provide further information and response. The following text has been added to the materials and methods section (Recruitment and Consent) L120 18th February and September 12th, 2022. With reference to the data being out of date, the authors respectfully disagree and draw the reviewer’s attention to the many aspects of the research process that need a completed data set prior to commencement (including birth outcomes). Reviewer 2 would be aware of the time required to conduct authentic co-production research with Indigenous peoples.

2. The Authors must explain how this work provides useful ideas for improvement related to the topic discussed, also in terms of Public Health and decisions that could be advanced by policy makers.

To the conclusion section, the following text has been added from L396. The results of this study translate to future public health research, practice, and policy. Future research and practice can be informed by using the NEST, a tested instrument that encompasses Aboriginal and Torres Strait Islander concepts of time providing a more accurate measure of nutritional intake. Policy makers have long been aware of the adverse impacts of women not birthing on country. These results complement previous research and highlight that the move away from Country forces fundamental changes to the woman’s nutritional intake.

Round 2

Reviewer 2 Report

Comments and Suggestions for Authors

The study remains largely dated, even with all the potentially plausible justifications.

Comments on the Quality of English Language

Minor editing of English language required.

Author Response

The study remains largely dated, even with all the potentially plausible justifications.

The authors acknowledge that the dates of data collection are clearly a priority for this reviewer, and would further expand on this point of concern. Participants were recruited across a wide gestational range, as well as postnatally. The final participant was recruited at 11 weeks gestation. She did not give birth until May 2023. At this time the data custodian at the hospital was approached with the application for release of medical records, which needed to follow the mandatory processes. Data cleaning and analysis followed, then the results needed to be reviewed with our Aboriginal and Torres Strait Islander partners, then authorship of the paper and related processes. The paper was written in as timely a manner as possible while adhering to the principles of Research in Aboriginal and Torres Strait Islander populations. The nature of the remote Aboriginal and Torres Strait Islander setting and sample remains unchanged; their dietary options, determinants and behaviours are not influenced by the western concept of time. No changes have been made in the manuscript.